# Isotopic evidence for initial coastal colonization and subsequent diversification in the human occupation of Wallacea

Patrick Roberts [1,2✉], Julien Louys [3], Jana Zech[1], Ceri Shipton [4,5], Shimona Kealy [4,5], Sofia Samper Carro[4,6], Stuart Hawkins[4,5], Clara Boulanger [5,7], Sara Marzo[1], Bianca Fiedler[1], Nicole Boivin[1], Mahirta[5,8], Ken Aplin[4,9] & Sue O'Connor [4,5✉]

The resource-poor, isolated islands of Wallacea have been considered a major adaptive obstacle for hominins expanding into Australasia. Archaeological evidence has hinted that coastal adaptations in *Homo sapiens* enabled rapid island dispersal and settlement; however, there has been no means to directly test this proposition. Here, we apply stable carbon and oxygen isotope analysis to human and faunal tooth enamel from six Late Pleistocene to Holocene archaeological sites across Wallacea. The results demonstrate that the earliest human forager found in the region *c.* 42,000 years ago made significant use of coastal resources prior to subsequent niche diversification shown for later individuals. We argue that our data provides clear insights into the huge adaptive flexibility of our species, including its ability to specialize in the use of varied environments, particularly in comparison to other hominin species known from Island Southeast Asia.

[1] Department of Archaeology, Max Planck Institute for the Science of Human History, 07745 Jena, Germany. [2] School of Social Science, The University of Queensland, St Lucia, QLD 4072, Australia. [3] Australian Research Centre for Human Evolution, Environmental Futures Research Institute, Griffith University, Nathan, QLD 4111, Australia. [4] School of Culture, History and Language, College of Asia and the Pacific, The Australian National University, Canberra, ACT 2600, Australia. [5] ARC Centre of Excellence for Australian Biodiversity and Heritage, Australian National University, Canberra, ACT 2600, Australia. [6] Centre d'Estudis del Patrimoni Arqueologic, Facultat de Lletres, Universitat Autònoma de Barcelona, 08193 Bellaterra, Spain. [7] Muséum National d'Histoire Naturelle, Département Homme et Environment, CNRS UMR 7194, Histoire Naturelle de l'Homme Préhistorique Paris, France. [8] Department of Archaeology, Universitas Gadjah Mada, Yogyakarta 55281, Indonesia. [9]Deceased: Ken Aplin. ✉email: roberts@shh.mpg.de; sue.oconnor@anu.edu.au

Recent, high-profile studies of symbolic material culture (e.g., ref. [1]), technological complexity (e.g., ref. [2]), fossil morphology and chronology (e.g., ref. [3]), and genetics[4] are demonstrating an increasingly complex and dynamic picture of the capacities and interactions of different hominin populations in the Late Pleistocene (126–12 ka), particularly in Asia. If we are to determine the 'uniqueness' of *Homo sapiens*, the last extant hominin on the face of the planet, it is becoming apparent that we must examine how its ecological adaptations differed from those of other members of the genus *Homo*[5,6]. It has been suggested that Late Pleistocene populations of *H. sapiens* expanding across the globe were able to not only flexibly exploit varied, and often extreme, environments—including deserts, tropical rainforests, high-altitude settings, and deep-sea maritime habitats—but also specialize in the occupation of them, enabling our species as a whole to proliferate even while local communities may sometimes have failed[6]. By contrast, earlier and contemporaneous *Homo* species expanding into Eurasia in the Early and Middle Pleistocene (2.6 Ma–126 ka) made generalized use of forest and grassland mosaics[7,8], potentially making them vulnerable to more extreme Late Pleistocene environmental changes (e.g., ref. [3]) and unable to survive on islands depauperate in large terrestrial fauna[9].

Testing this hypothesis is particularly timely given recent finds that imply other hominin species may have ventured into challenging adaptive settings[4,10]. Wallacea provides an ideal 'island laboratory' setting in which to do so in the increasingly palaeoanthropologically significant Southeast Asian region. Wallacea is an isolated series of islands that was never connected to the neighboring Pleistocene landmasses of Sunda or Sahul, necessitating water crossings to reach[9,11–15]. These islands have been hypothesized as hosting depauperate island forest environments, lacking in reliable terrestrial protein and carbohydrate resources[13,16,17]. Significantly, while these islands are home to some of the earliest firm evidence for *H. sapiens* east of Africa and the Middle East c. 45 ka (refs. [13,18,19]), fossil and artifact finds have also suggested the presence of earlier members of the genus *Homo* on the island of Flores from ~1 Ma (refs. [20,21]), Luzon from 0.7 Ma (ref. [22]), and Sulawesi from ~0.2 Ma (ref. [23]). Although zooarchaeological records have provided some insights into the ecological niches of different hominin populations in Wallacea[9,24], more direct assessments of overall hominin resource reliance and palaeoenvironmental change in the region have been lacking.

In this paper, we examine the adaptations of the earliest known fossil members of our species in Wallacea by means of isotopic analysis of archaeological human tooth enamel from two islands (Timor and Alor; Fig. 1). Timor has yielded the earliest dated material culture and fossil evidence for *H. sapiens* in Wallacea at the sites of Asitau Kuru (formerly Jerimalai) and Laili[18,19]. At the former, faunal remains and cultural artifacts suggest Late Pleistocene human reliance on marine shellfish and fish, obtained in part through offshore fishing[13]. Laili also provides evidence for early reliance on marine resources[18]. This stands in contrast to the generalized mixed grassland and woodland adaptations associated with other hominins in the region[9,24,25]. However, human reliance on pelagic fishing at Asitau Kuru has been questioned[26]. Moreover, there remains the possibility that giant rat taxa, with proposed preferences for closed forest environments and an adult body weight of up to 6 kg, represented significant food resources; and they have been identified in early coastal and inland archaeological contexts in Alor and Timor (e.g., ref. [27]). Insights into the environments present on Wallacea, as well as human reliance on different ecosystems is difficult to resolve

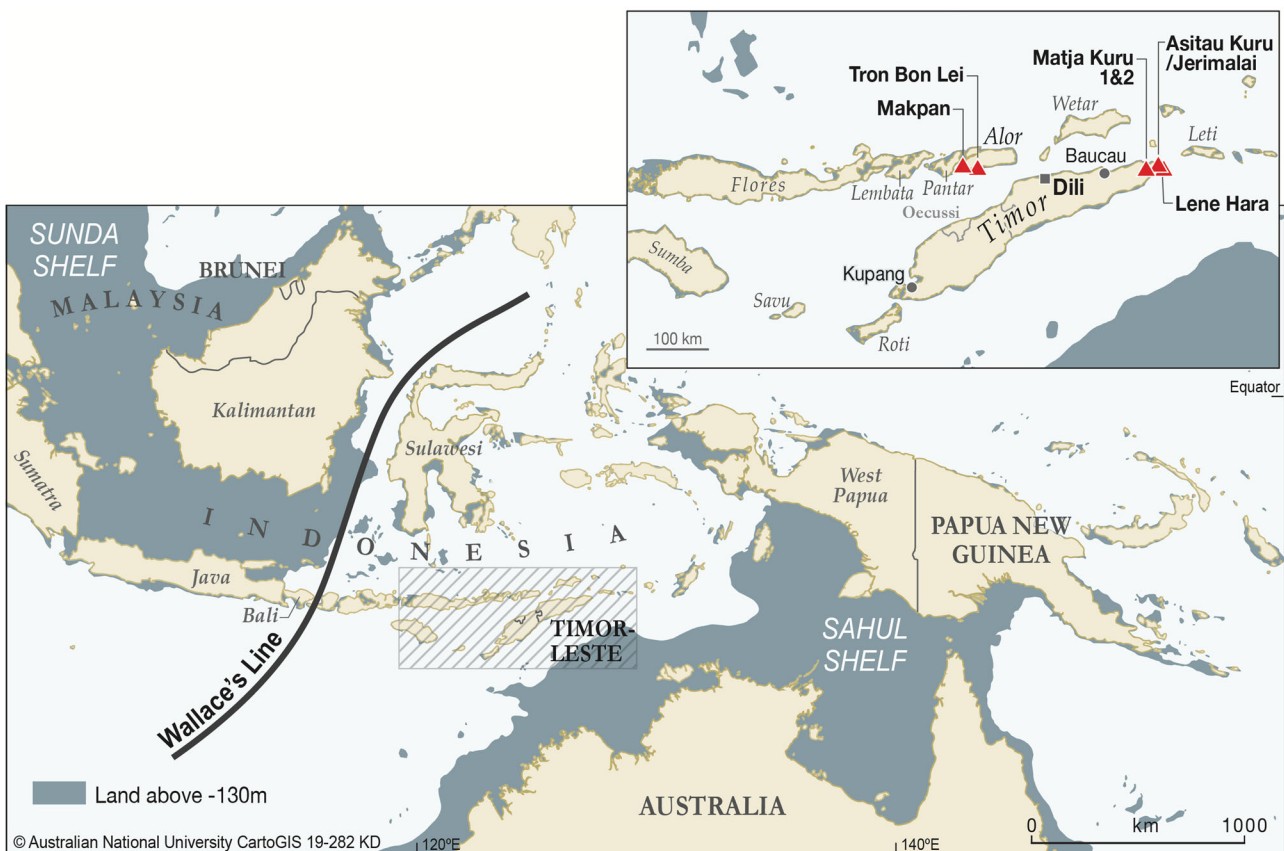

**Fig. 1 Maps showing the location of the studied sites within Wallacea.** Asitau Kuru, Lene Hara, Matja Kuru 1 and 2 (Timor), Makpan, and Tron Bon Lei (Alor).

using traditional zooarchaeological methods alone due to preservation biases and the role of nonhuman predators in site taphonomy (e.g., ref. [28]).

Here, we apply stable carbon ($\delta^{13}C$) and oxygen ($\delta^{18}O$) isotope analyses to human and faunal tooth enamel from six Late Pleistocene/Holocene archaeological sequences (Fig. 1) on Timor and Alor, in order to determine the varying reliance of early human colonisers of Wallacea on tropical forest and terrestrial versus marine resources. Stable carbon isotope analysis of faunal (including hominin) tooth enamel in tropical regions has been used to assess the proportion of $C_3$-dominated woodland/forest and $C_4$ grassland biomass in diets[29–31]. In regions such as Pleistocene Wallacea, where some researchers have suggested that tropical forests dominated terrestrial environments[32], with grasslands considered largely absent, the most significant driver of terrestrial stable carbon isotope variation will be the canopy effect, whereby low light and respired $CO_2$ cause forest-dwelling plant biomass and its consumers to have more negative $\delta^{13}C$ values than their counterparts in more open habitats[30,31]. Meanwhile, marine producer biomass has higher $\delta^{13}C$ than all $C_3$ terrestrial plants[33,34], enabling marine consumers to be distinguished from terrestrial $C_3$ consumers[35]. Based on research done in East Africa[30], Sri Lanka[31], and Japan[35], including extensive modern studies[30], we expect preindustrial humans relying completely on tropical forest, open $C_3$ resources, and marine resources to have tooth enamel $\delta^{13}C$ values of c. −14‰, c. −11‰, and c. −4‰, respectively.

Stable oxygen isotope ($\delta^{18}O$) measurements from animal tooth enamel provide additional paleoecological information about water and food, and have also been argued to distinguish terrestrial from marine consumers[36]. Based on existing, published, and available chronological information, the Late Pleistocene–Holocene deposits of Asitau Kuru, Matja Kuru 1 and 2, Lene Hara, Makpan, and Tron Bon Lei provide a unique suite of human and associated faunal samples spanning the earliest fossil appearance of *H. sapiens* in Wallacea, through the Last Glacial Maximum, and across the Terminal Pleistocene–Holocene transition[18,19]. They also cover both coastal and hinterland habitats (Fig. 1). Ample terrestrial and marine animal remains also allow us to build the first detailed paleoecological and palaeoenvironmental records for Pleistocene Wallacea and test assumptions in relation to: (1) pure $C_3$ terrestrial environments on Timor and Alor in the past; (2) the $\delta^{13}C$ distinction between available terrestrial and marine resources; and (3) environmental shifts across the Pleistocene–Holocene boundary proposed elsewhere in Southeast Asia (e.g.,[5,37]). The preservation of a subsection of the analysed tooth enamel samples was also checked using Fourier transform infrared spectroscopy (FTIR) as per Roberts et al.[31,38].

Our extensive faunal baseline demonstrates that terrestrial and marine environments can be clearly distinguished isotopically in Wallacea on the basis of stable isotope analysis of fossil tooth enamel. We show that a tooth of the earliest preserved *H. sapiens* fossil found from the region c. 42–39,000 years ago shows that this individual made significant use of coastal resources. From 20,000 years ago, human populations show an increasing reliance on interior, terrestrial environments on the islands of both Timor and Alor at a time of increasing forest expansion in Island Southeast Asia more generally, though some individuals continue to intensively use marine resources. We argue that our data further demonstrates the huge adaptive flexibility of our species, acutely visible as it rapidly and persistently colonized Wallacean environments. Its ability to specialize in the use of more extreme environments seems to stand in contrast to other hominin species known from Island Southeast Asia based on current evidence.

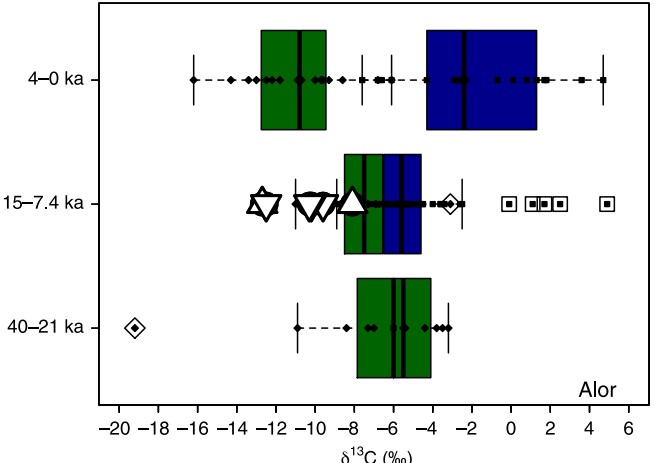

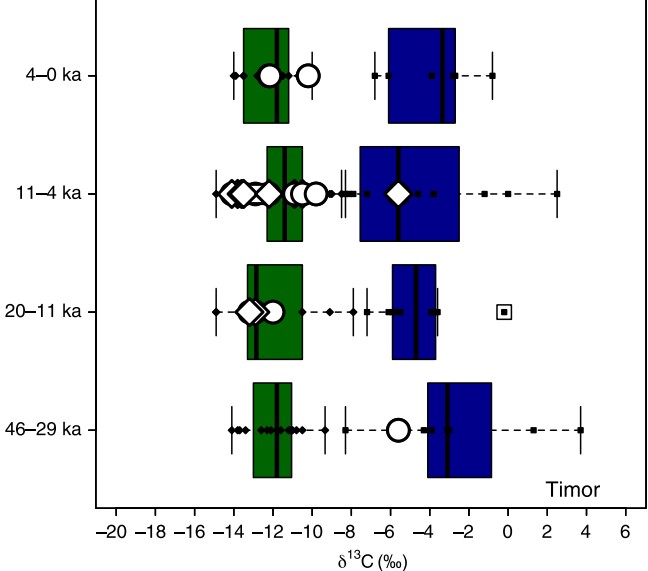

**Fig. 2 $\delta^{13}C$ measurements for human and faunal tooth enamel from the islands of Alor and Timor analyzed in this study.** Data shown by phases developed on the basis of existing and published stratigraphic, and chronological information (Supplementary Note 1). Boxplots (showing median and interquartile range—with outliers indicated) of terrestrial fauna are shown in green (black diamond points) and those of marine fauna in blue (black square points). Human samples are shown as white symbols depending on site (circle, Asitau Kuru; diamond, Matja Kuru 1 and 2; square, Lene Hara; triangle, Makpan; and inverted triangle, Tron Bon Lei). Source data for Fig. 2 can be found in the accompanying Source Data file in Table 1 (Fig. 2).

## Results

**Sites, samples, and chronology**. The detailed stratigraphic and chronological information for the six sites studied (Supplementary Note 1) has enabled division of the human and faunal samples into occupation phases at each site (Figs. 2–6). To display and compare our data on a broader scale, we have also divided the human and faunal data into broader island phases of occupation for Timor and Alor, respectively, based on the stratigraphic and chronometric information (Supplementary Note 1, Supplementary Tables 6 and 7, Fig. 2). For the sites of Asitau Kuru, Lene Hara, Matja Kuru 1, and Matja Kuru 2 on Timor, this system includes four broad phases: a Late Pleistocene pre-LGM phase (46,000–29,000 years ago), a Terminal Pleistocene phase (20,000–11,001 years ago), an Early and Middle Holocene phase (11,000–4001 years ago), and a Late Holocene Neolithic phase (4000–0 years ago). For the sites of

Makpan and Tron Bon Lei on Alor, the phasing system includes three broad phases: a Late Pleistocene pre-LGM phase (40,000–21,000 years ago), a Terminal Pleistocene to Middle Holocene phase (15,000–7400 years ago), and a Late Holocene Neolithic phase (4000–0 years ago). The specific associated dates for the human samples within these broader phases are discussed in the main text where appropriate.

**Stable isotope analysis of archaeological tooth enamel.** Faunal $\delta^{13}C$ from the Late Pleistocene–Holocene sequences of Asitau Kuru, Matja Kuru 2, Makpan, and Tron Bon Lei (Figs. 2–6) show a $\delta^{13}C$ division in terms of terrestrial and marine fauna (Fig. 2, Supplementary Figs. 13 and 14, Supplementary Data 1). For the island of Timor, terrestrial fauna recovered from the coastal site of Asitau Kuru (Fig. 3) and inland site of Matja Kuru 2 (Fig. 4) have $\delta^{13}C$ ranges of −14.9 to −7.9‰ (mean = −11.6 ± 1.8‰) and −14.9 to −9.0‰ (mean = −12.0 ± 1.4‰), respectively. By contrast, the marine fauna from Asitau Kuru (Fig. 3) has a $\delta^{13}C$ range of −8.3 to 3.7‰ (mean = −4.1 ± 3.0 ‰). The situation is slightly more complex on Alor (Fig. 2) where terrestrial fauna from Makpan (Fig. 5) has $\delta^{13}C$ ranging from −19.2‰ to −3.1‰ (mean = −8.7 ± 3.5‰), and marine fauna from Makpan (Fig. 5) and Tron Bon Lei (Fig. 6) have $\delta^{13}C$ ranging from −6.4 to 4.9‰ (mean = −3.2 ± 2.9‰) and −8.9 to 4.7‰ (mean = −5.4 ± 3.5‰), respectively.

A Shapiro–Wilk test indicated that the $\delta^{13}C$ ($p = < 0.05$) and $\delta^{18}O$ ($p = < 0.05$) of the entire faunal dataset ($n = 223$) were non-normally distributed. Mann–Whitney–Wilcoxon tests demonstrated the $\delta^{13}C$ ($W = 9257$, $p = < 0.05$) and $\delta^{18}O$ ($W = 5264$, $p = < 0.05$) of fauna to be significantly different between Timor and Alor. Consequently, the faunal $\delta^{13}C$ and $\delta^{18}O$ datasets of the islands (Timor $n = 111$; Alor $n = 112$) were separated for subsequent analyses unless otherwise specified. For both Timor and Alor, Shapiro–Wilk tests found the resulting $\delta^{13}C$ and $\delta^{18}O$ datasets for each island to be non-normally distributed ($p = < 0.05$). Mann–Whitney–Wilcoxon tests found marine and terrestrial fauna to be significantly different in terms of $\delta^{13}C$ on both Timor ($W = 2657$, $p = < 0.05$) and Alor ($W = 2423$, $p = < 0.05$). No difference was found between terrestrial and marine faunal groups in terms of $\delta^{18}O$ on either Timor ($W = 1507$, $p = > 0.05$) or Alor ($W = 1375$, $p = > 0.05$).

In terms of terrestrial palaeoenvironmental conditions and changes in Late Pleistocene–Holocene Wallacea, our data directly confirms the $C_3$ forest–woodland preferences for now-extinct giant rat taxa on both Timor and Alor (Figs. 3–5; as per ref. [27]). Separation of the terrestrial fauna dataset from Timor ($n = 76$) and Kruskal–Wallis analysis shows there to be no significant $\delta^{13}C$ differences by phase on this island (Kruskal–Wallis chi-squared = 3.157, df = 3, $p = > 0.05$). Although significant differences in $\delta^{18}O$ were noted between phases (Kruskal–Wallis chi-squared = 13.647, df = 3, $p = <0.05$), pairwise comparison failed to draw out any specific differences (>0.05) (Supplementary Table 8). For the Alor terrestrial faunal dataset ($n = 42$), the situation is more complicated. Here, a Kruskal–Wallis test (Kruskal–Wallis chi-squared = 14.386, df = 2, $p = <0.05$) followed by pairwise comparison demonstrated significant $\delta^{13}C$ differences between phases A (40,000–21,000 years ago) and C (4000–0 years ago), and B (15,000–7400 years ago) and C (4000–0 years ago; Supplementary Table 9). No significant differences were found for $\delta^{18}O$ (Kruskal–Wallis chi-squared = 2.321, df = 2, $p = > 0.05$).

The difference between the islands is driven by the fact that in the earliest phase of occupation on Alor (40–21,000 cal. years BP) there is an overlap between terrestrial and marine fauna in $\delta^{13}C$ (Figs. 2 and 5). From this point, the $\delta^{13}C$ of terrestrial fauna declines through the phases, with terrestrial and marine fauna

becoming obviously different between 15,000–7400 cal. years BP and 4000–0 cal. years BP (Fig. 2). These results suggest that $C_4$ resources may have been available to some terrestrial fauna in the earliest phase of human occupation on Alor, with their presence declining through time. However, it is also possible that elevated rat $\delta^{13}C$ could be a product of early access to marine resources[39]. Finally, separation of the combined dataset of marine fauna for Timor and Alor ($n = 105$), and Kruskal–Wallis analysis and pairwise comparison (Kruskal–Wallis chi-squared = 17.975, df = 8, $p = < 0.05$) showed significant $\delta^{13}C$ differences between reef taxa such as Balistidae (Triggerfishes) and more wide-ranging taxa, such as *Scaridae* (Parrotfishes) (Supplementary Table 10), indicating the potential utility of isotopic analysis of teeth to distinguish fish from different marine niches (see also ref. [40]).

Our large, robust faunal baseline enables the long-term ecological niches of Late Pleistocene/Holocene human foragers to be directly determined for Asitau Kuru, Matja Kuru 2, Makpan, and Tron Bon Lei, as well as the additional sites of Lene Hara and Matja Kuru 1 where only human samples were available (Figs. 2–6; $n = 26$). Sampled human $\delta^{13}C$ and $\delta^{18}O$ ranges between −14.1 and −5.6‰ and −6.2 to −3.2‰, respectively (Fig. 2, Supplementary Data 2). The earliest human sample in the study, and the earliest recovered from Wallacea, from context B63 at Asitau Kuru, dated to c. 42,440–38,853 cal. years BP, has a $\delta^{13}C$ value of −5.6‰ (Fig. 3). This is indicative of a high reliance on marine resources, given the lack of any evidence for $C_4$ resources in this part of Timor at this time and zooarchaeological evidence for abundant marine resources (Fig. 2). The majority of the remaining humans sampled from the Terminal Pleistocene and Holocene contexts of Asitau Kuru (Fig. 3), Lene Hara (Fig. 3), Matja Kuru 1 and 2 (Fig. 4), and the site of Makpan (Fig. 5) on Alor, have $\delta^{13}C$ values between −14.1‰ and −9.6‰, indicating human reliance on a mixture of terrestrial tropical forest resources and more open $C_3$ environments (Fig. 2).

Indeed, Fig. 6 demonstrates that even the $\delta^{13}C$ values of humans excavated from Tron Bon Lei (Supplementary Note 1), with an assumed economic and cultural reliance on marine resources, have a range of between −12.5‰ and −9.6‰. With perhaps the exception of the individual from square C with a value of −9.6‰, who likely demonstrates some contribution of marine or $C_4$ resources, this shows that terrestrial $C_3$ resources made up the majority of the diets of these individuals. Nevertheless, it is clear that two Terminal Pleistocene/Holocene individuals at Makpan, Alor (−8.1‰; 15,000−11,000 cal. years BP; Fig. 5), and Matja Kuru 2, Timor (−5.6‰; Fig. 4; 11,000–4000 cal. years BP) incorporated a significant proportion of $C_4$/marine and marine resources into their diets, respectively, based on associated faunal data (Fig. 2), suggesting that diets were diverse between individuals and societies during the Terminal Pleistocene and Holocene.

**Fourier transform infrared spectroscopy.** Full results of the infrared indices of samples subjected to FTIR analysis are shown in Supplementary Data 3. All of the fossil and modern enamel samples displayed classic enamel FTIR spectra (Supplementary Fig. 15). No additional bands from secondary carbonate (e.g. calcite at 710 cm$^{-1}$; ref. [41]) were observed in the spectra of the fossil samples (Supplementary Fig. 15). Boxplots of API, BPI, WAMPI, PCI, and BAI for the groups of 'Modern', 'Fossil Human', 'Fossil Terrestrial Fauna', and 'Fossil Marine Fauna' are shown in Supplementary Fig. 16. Broadly, there are minimal changes between the groups with fossil samples perhaps having marginally lower API, slightly higher BPI, and higher BAI than the 'Modern' sample (Supplementary Fig. 16). Analysis of variance (ANOVA) and post-hoc Tukey pairwise comparisons

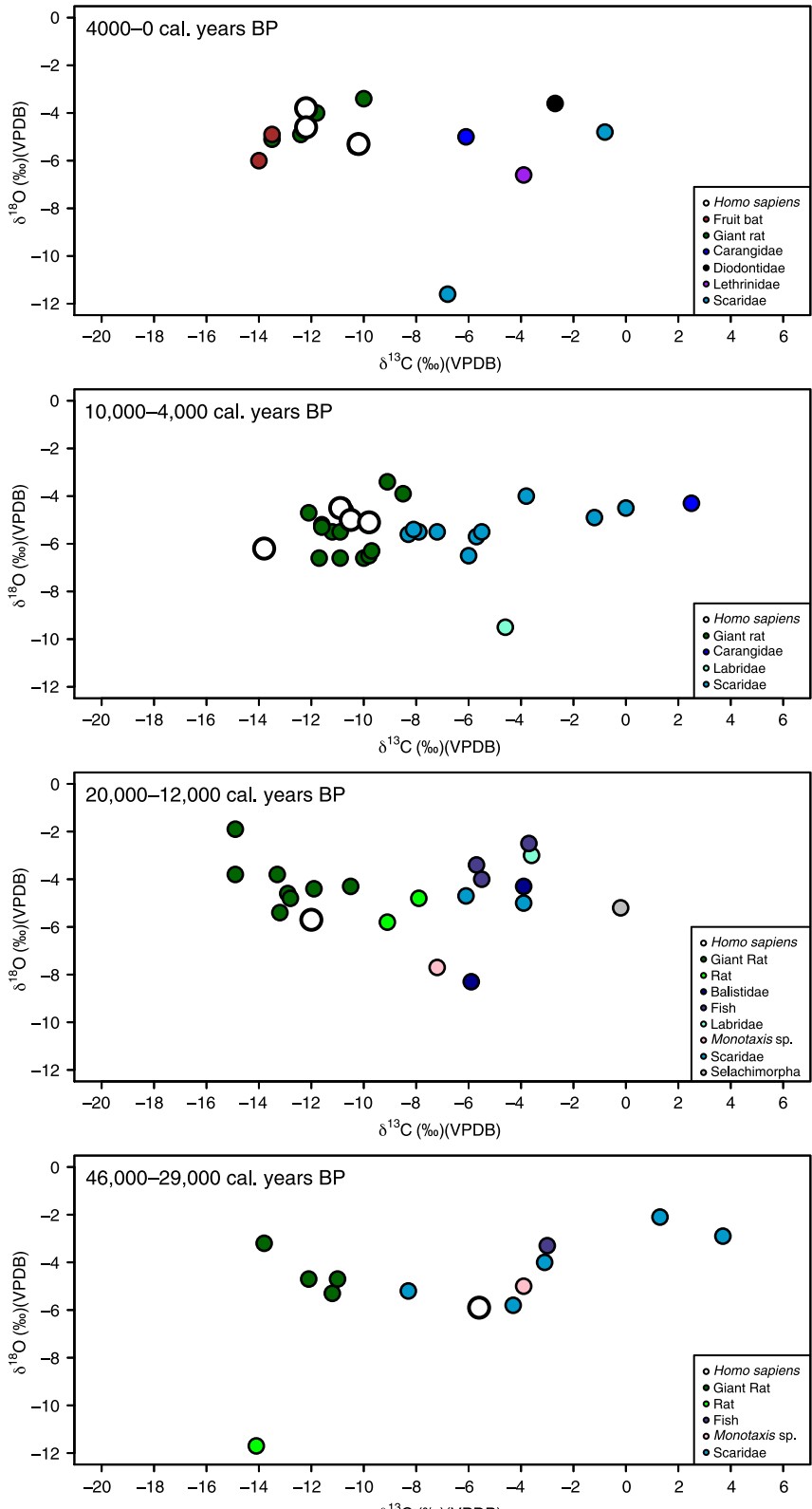

**Fig. 3 Isotope data from Asitau Kuru (Timor).** Stable carbon ($\delta^{13}$C) and oxygen ($\delta^{18}$O) isotope data from terrestrial and marine faunal tooth enamel samples, with human samples shown as white circles, displayed by site phases (see Supplementary Note 1).

support this, finding no significant differences in A-site carbonation ($F(3,55) = 1.194$, $p > 0.05$), B-site carbonation ($F(3,55) = 1.387$, $p > 0.05$), PCI ($F(3,55) = 1.845$, $p > 0.05$), or WAMPI ($F(3,55) = 1.245$, $p > 0.05$) between the different sample groups. By contrast, BAI does show a difference between the groups

($F(3,55) = 6.232$, $p < 0.05$), with 'Fossil Marine Fauna' and 'Fossil Terrestrial Fauna' being significantly different from 'Modern' samples (Supplementary Table 11).

Poor preservation of skeletal material has been suggested elsewhere in tropical rainforest environments, though such

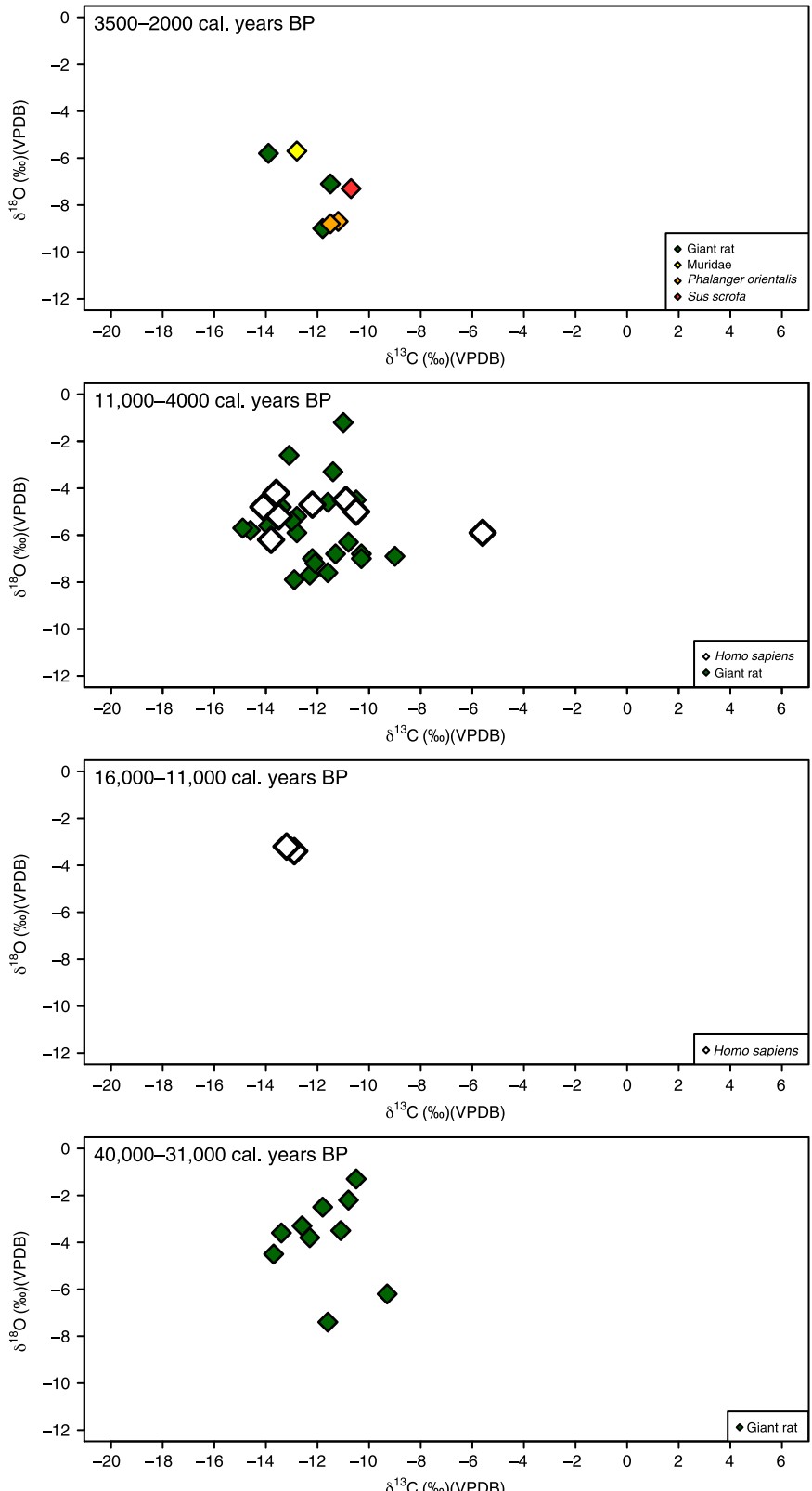

**Fig. 4 Isotope data from Matja Kuru 1 and 2 (Timor).** Stable carbon ($\delta^{13}$C) and oxygen ($\delta^{18}$O) isotope data from terrestrial faunal tooth enamel samples from Matja Kuru 2 (Timor), with human samples shown as white diamonds, displayed by site phases (see Supplementary Note 1). Regional phasing has been used for the 11,000–4000 cal. years BP grouping so individuals from Matja Kuru 1 and 2 can be combined. The grouping of 16,000–11,000 cal. years BP is only represented by two human individuals from Matja Kuru 1 as there is no occupation at Matja Kuru 2 at this time (see Supplementary Note 1).

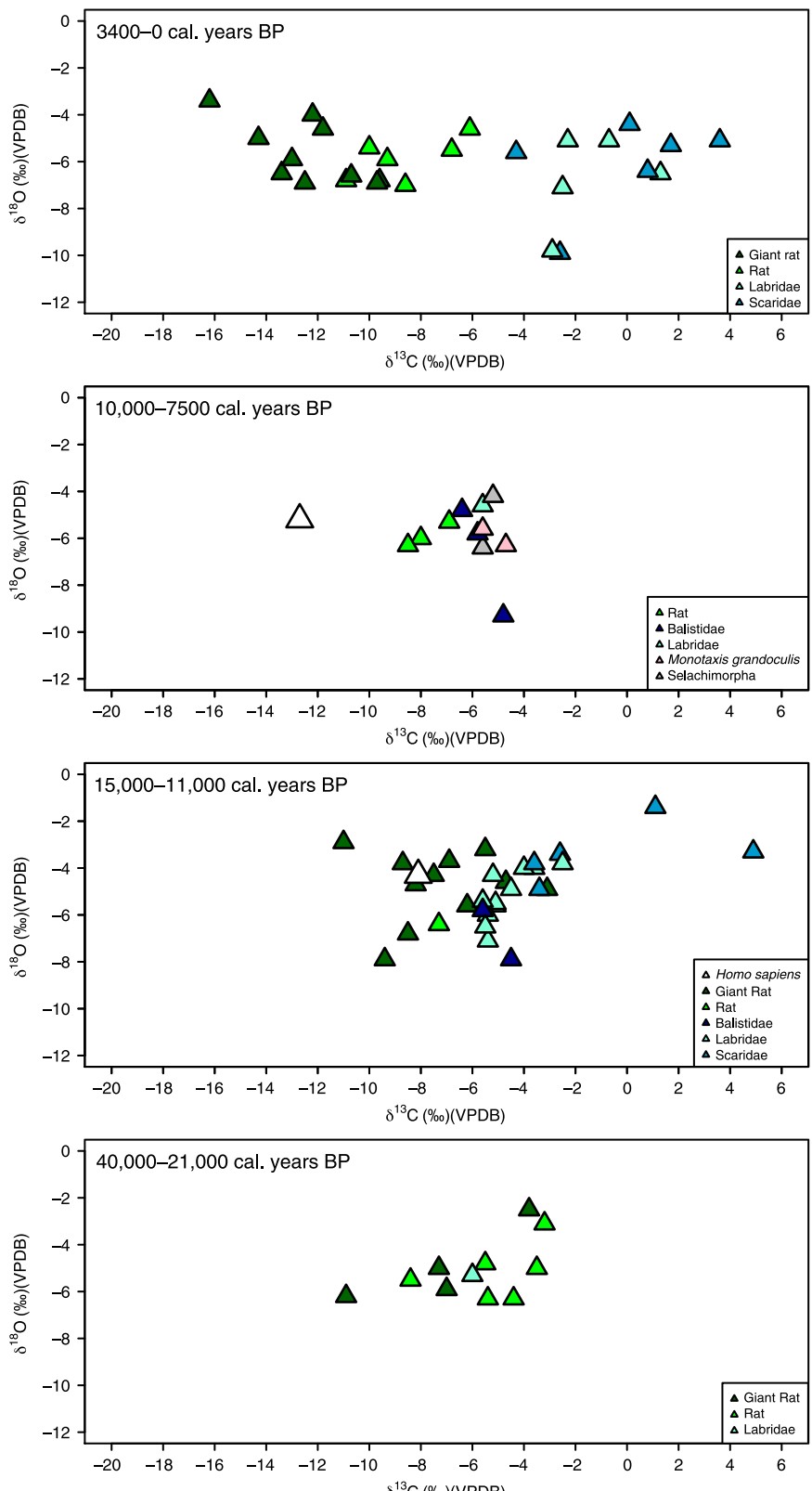

**Fig. 5 Isotope data from Makpan (Alor).** Stable carbon ($\delta^{13}C$) and oxygen ($\delta^{18}O$) isotope data from terrestrial and marine faunal tooth enamel samples, with human samples shown as white triangles, displayed by site phases (see Supplementary Note 1).

comments have mainly focused on organic bone material[42]. The fossil enamel FTIR spectra produced here are virtually indistinguishable from modern spectra (Supplementary Figs. 15 and 16). The precipitation of carbonate minerals is likely to be more

of a problem in the context of more porous materials, such as bone[43,44]. Subtle differences noted in faunal enamel apatite during fossilization noted here, including increased BAI and decreased A-carbonate on phosphate index, have also been

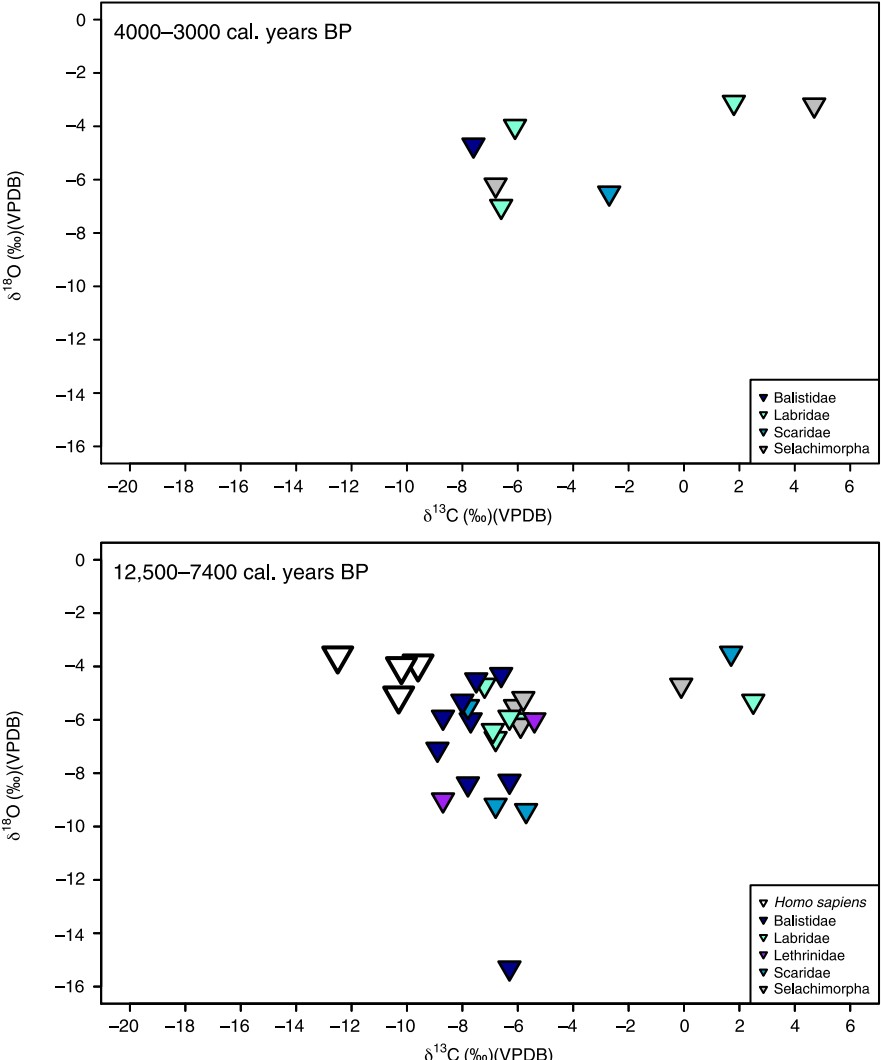

**Fig. 6 Isotope data from Tron Bon Lei (Alor).** Stable carbon ($\delta^{13}C$) and oxygen ($\delta^{18}O$) isotope data from terrestrial and marine faunal tooth enamel samples, with human samples shown as white inverted triangles, displayed by site phases (see Supplementary Note 1).

## Discussion

Our stable isotope data allow us to directly assess the ecological reliance on different categories of resources that accompanied our species' arrival and subsequent settling of Wallacea. The earliest humans to arrive in this part of the world seemingly specialized in the use of coastal resources. While we cannot currently distinguish between pelagic and other forms of offshore resource use[13], and our sample size from this early period is limited, we can be confident that the $\delta^{13}C$ value for this individual indicates a reliance on marine resources. Following 20,000 cal. years BP, a clear diversification in human resource use across Wallacea emerges. While some coastal reliance is indicated by one individual at Matja Kuru 2, and perhaps also Makpan, most individuals

demonstrated in other studies, including within the tropics of South Asia, and have been argued to be a product of the reduction in organic material within the apatite matrix through time[31,38,45]. Such change is not considered to have major impacts on overall enamel apatite structure or stable carbon and oxygen isotope measurements from enamel[45], as further suggested by the preservation of expected ecological differences in the enamel here and elsewhere[29,31].

demonstrate broader use of interior environments, including closed tropical forest habitats. This may be considered surprising, particularly given the ongoing presence of fish and shellfish[9], the symbolic burial of an individual at Tron Bon Lei with fishhooks (Supplementary Note 1), as well as archaeological evidence for increased transfer of material culture between islands from this time[46,47]. However, tooth enamel $\delta^{13}C$ reflects the whole diet of an individual, and our data highlights the necessity of paying more attention to the contribution of plant (and terrestrial animal) resources to human diets on tropical islands, particularly as those populations become more established—something also recently urged even for the study of the later Lapita expansion in the Pacific[48,49].

This study presents the first detailed palaeoenvironmental information for Late Pleistocene–Holocene Wallacea, directly associated with records of human behavior. Our data indicates that, on Timor, tropical forest environments remained prevalent throughout the past 45,000 years, only decreasing during the Late Holocene with the arrival of human-induced deforestation during the Iron Age[27,50]. There is no evidence for the presence of $C_4$ grassland environments in the vicinity of any of the sites studied. By contrast, on Alor, $C_4$ resources may have been available to some small mammals, and presumably also humans, during the earliest

period of occupation (40,000–21,000 years ago). From this point onward, these resources disappear as tropical forest environments expanded across the Terminal Pleistocene–Holocene boundary. While it remains possible that elevated $\delta^{13}C$ for some fauna in the earlier phase represents consumption of marine resources, this palaeoenvironmental pattern of increasing tropical forests during the Terminal Pleistocene has also been documented elsewhere in both Island and Mainland Southeast Asia[5]. In these cases, the expansion of tropical forest during the Terminal Pleistocene and Early Holocene has also been associated with increasingly specialized human hunting of arboreal and semi-arboreal mammals and the use of tropical forest plants[5].

Our isotopic evidence supports human colonization models of Wallacea and Australia that suggest a rapid, initial coastal colonization, followed by later inland settlement[12,14], at least with regards to the sites studied here. This mode of colonization is distinct from isotopic and material evidence from the Wet Zone rainforests of Late Pleistocene Sri Lanka[31,51] and archaeological evidence from the Niah Caves in Borneo[52] that indicate dedicated, specialized tropical forest foraging by early human populations in these regions from 45,000 years ago. This further highlights the potential role of sophisticated seafaring in the human colonization of eastern Wallacea and Australasia[12,13,53]. A later, increased focus on terrestrial resources or nearshore coastal resources during the Terminal Pleistocene and Holocene has also been argued on the basis of zooarchaeological evidence from Timor, and elsewhere in Island Southeast Asia[18,54]. This subsistence pattern occurs alongside an increase in occupation intensity across Wallacea from the Terminal Pleistocene, as well as an increase in formalized long-distance trading networks, and is likely representative of exchange between settled groups in the region[46,47].

The adaptive flexibility visible in the colonization of almost all of the Earth's continents by our species in the Late Pleistocene stands in stark contrast to the adaptations of other hominin species[3,6,9,24]. While there is some evidence that other hominins may have made water crossings[20–23] or ventured into high-altitude environments[10], where present, existing zooarchaeological/paleontological and palaeoenvironmental evidence suggests a general, albeit diverse, focus on mixed grassland and woodland environments, with dispersals and contractions often reliant on climatically driven environmental change[24,55]. Future isotopic analysis and more detailed zooarchaeological work are required to test this distinction, both in Wallacea and beyond. However, there is clear evidence that different populations of *H. sapiens* were able to specialize in a variety of extreme environments even as our species as a whole generalized in the use of multiple settings[6]. This flexibility, perhaps supported by unique capacities of innovation, technological sophistication, and social communication (e.g., ref. [56]), enabled adaptation to a variety of conditions, not just through space but also through time, that would eventually leave us the last hominin standing.

## Methods

**Sites, samples, and chronology.** We sampled available human and animal teeth from the Late Pleistocene–Holocene deposits of Asitau Kuru, Matja Kuru 1 and 2, and Lene Hara on the island of Timor and Makpan and Tron Bon Lei on the island of Alor. Existing, including published, stratigraphic, and chronological information was compiled, in order to enable the placement of samples in their proper context. All of the available dates used, stratigraphic information, and accompanying archaeological finds can be found in Supplementary Note 1, Supplementary Figs. 1–12, and Supplementary Tables 1–7. The available samples were selected on the basis of occupation phases noted and published for each archaeological site (Supplementary Note 1, Supplementary Tables 1–7). Based on these established sequences, samples were also grouped into an overall 'phasing' for each of the islands of Timor and Alor to provide larger sample sizes for the broader evaluation of adaptive context for the arrival of humans on each island, hypothesized palaeoenvironmental changes across the Terminal Pleistocene/Holocene boundary,

and changes following the arrival of 'Neolithic' material culture during the Holocene (Supplementary Note 1, Supplementary Tables 6 and 7).

Although bone collagen is frequently the primary tissue used in the isotopic determination of ancient diets, it is often degraded and nearly impossible to extract from archaeological remains in the humid tropics, particularly those dating back to the Pleistocene[57,58]. This is the case for the sites studied here. By contrast, tooth enamel consists primarily of an inorganic fraction extremely resilient to postmortem diagenesis[59], meaning that it is the fossil tissue of choice for tropical dietary reconstruction[58,60]. Tooth identification and analyses were conducted at the Australian National University (ANU). Fish, reptile, and non-murid mammal identifications were facilitated through comparisons with specimens from the ANU Archaeology and Natural History Osteology Laboratory reference collection. Murid identifications were facilitated through comparisons with archaeological and fossil specimens collected from previous ANU expeditions, material held in KA's private collections, the Australian National Wildlife Collection of the Commonwealth Scientific and Industrial Research Organisation National Facilities and Collections, and descriptions and illustrations in Aplin and Helgen[61] and Glover[62].

Tooth enamel records an isotopic dietary signature for the period of enamel formation that will vary depending on the species and tooth sampled. For humans, the longest isotopic signature is provided by third molars that can develop any time between 7 and 13 years of age, and the mid to late teenage years[63,64]. While collagen $\delta^{13}C$ is also biased toward protein components of the diet[65], tooth enamel $\delta^{13}C$ reflects that of the whole diet during formation. Due to the rarity of fossil human remains in Pleistocene archaeological sites, we sampled all of the available human teeth present (Supplementary Data 2). The result is one of the largest collections of human tooth enamel data that spans the Late Pleistocene/Holocene (see refs. [31,60] for comparison in Sri Lanka) and the first for Island Southeast Asia. While some of these teeth come from skeletons that have been aged and sexed, the majority are loose teeth. Subtle variations in $\delta^{13}C$ and $\delta^{18}O$ between teeth could occur as a result, particularly for teeth formed during 'weaning'[66]. However, overall this minor 'noise' will not hinder the testing of the major $\delta^{13}C$ distinctions between closed canopy $C_3$ resources, $C_3$ open settings, $C_4$ resources, and marine resources.

**Stable isotope analysis of archaeological tooth enamel.** All teeth or teeth fragments were cleaned using air-abrasion to remove any adhering external material. Enamel powder for bulk analysis was obtained using gentle abrasion with a diamond-tipped drill along the full length of the buccal surface, in order to ensure a representative measurement for the entire period of enamel formation. All enamel powder was pretreated to remove organic or secondary carbonate contaminates. This method followed established protocols that have been applied elsewhere to Pleistocene tooth enamel in the tropics, where it has proven to be effective[31,60,67,68], enabling future comparison between datasets. Samples were washed in 1.5% sodium hypochlorite for 60 min, followed by three rinses in purified $H_2O$ and centrifuging, before 0.1 M acetic acid was added for 10 min, followed by another three rinses in purified $H_2O$. Samples were then lyophilized for 24 h.

Following reaction with 100% phosphoric acid, gases evolved from the samples were measured by stable carbon and oxygen isotope analysis using a Thermo Gas Bench 2 connected to a Thermo Delta V Advantage Mass Spectrometer at MPI-SHH. $\delta^{13}C$ and $\delta^{18}O$ values were compared against International Standards (IAEA-603 ($\delta^{13}C = 2.5$; $\delta^{18}O = -2.4$); IAEA-CO-8 ($\delta^{13}C = -5.8$; $\delta^{18}O = -22.7$); USGS44 ($\delta^{13}C = -42.2$)) and in-house standard (MERCK ($\delta^{13}C = -41.3$; $\delta^{18}O = -14.4$)) using Isodat 3.0 software from Thermo Electron Corporation. Replicate analysis of MERCK carbonate standards suggests that machine measurement error is c. ±0.1‰ for $\delta^{13}C$ and ±0.2‰ for $\delta^{18}O$. While each sample was measured only once to preserve material for future analyses, as is standard for this approach, overall measurement precision and reproducibility for tooth enamel samples on this machine setup was studied through the measurement of repeat extracts from an in-house bovid tooth enamel standard ($n = 20$; $\delta^{13}C = -12.4 \pm 0.2$‰; $\delta^{18}O = -8.0 \pm 0.3$‰).

**Fourier transform infrared spectroscopy.** While reliable $\delta^{13}C$- and $\delta^{18}O$-based dietary and environmental indicators have been demonstrated across millions of years[29], protocols to check the structural preservation of fossil tooth enamel samples remain important (see ref. [69]). This is particularly the case in tropical forest environments with ion-rich soils and high hydrological activity. One means to check enamel preservation is the application of FTIR, which absorbs radiation at discrete vibrational frequencies related to the presence and crystallographic environment of key functional groups (see refs. [45,70,71]). The polyatomic ions of interest are phosphates ($PO_4^{3-}$), carbonates ($CO_3^{2-}$), and hydroxyl groups ($OH^-$). The observed absorbance bands of enamel can be ascribed to the internal vibrations of these molecular groups[41,72] (Supplementary Table 12).

We use the empirical indices from Sponheimer and Lee-Thorp[70], and Roche et al.[45] to characterize the crystal-chemical properties of enamel bioapatite (Supplementary Table 12). The possible presence of calcite was assessed in all samples by checking for a peak at 710 cm$^{-1}$ (refs. [41,70]). 14 'Fossil Human', 15 'Fossil Terrestrial Fauna', and 15 'Fossil Marine Fauna' samples were subjected to FTIR analysis following pretreatment, in order to determine the remaining potential for diagenetic structural and compositional modification of enamel after pretreatment. Samples were randomly selected to cover a variety of the temporal phases and all of the sites studied (Supplementary Data 3). The fossil spectra were compared to those

available for 15 modern primate and cervid samples, and historical (late nineteenth and early twentieth century) human enamel samples ('Modern'), from populations living in tropical forest environments in Sri Lanka already published by Roberts et al.[73] (Supplementary Data 3).

For all samples, powdered enamel was analysed between 400 and 4,000 cm$^{-1}$ by FTIR with Attenuated Total Reflectance (FTIR-ATR—Bruker Vertex 70 v) using the OPUS 8.5 software from Bruker. Each sample was measured three times. The background was subtracted and a baseline correction was carried out using the OPUS 8.5 software from Bruker. The baselines of the spectra were normalized and all three spectra of each sample were averaged before calculation of the various infrared indices. To ensure better reproducibility of the measurements, only spectra with a minimum absorbance of 0.06 for the highest phosphate band at ~1035 cm$^{-1}$ were taken into account. The reproducibilities of the indices BPI, API, BAI, and PCI are ±0.01, ±0.007, ±0.1, and ±0.1, respectively.

**Statistical analysis**. All δ$^{13}$C and δ$^{18}$O datasets were tested for normality using the Shapiro–Wilk test and histogram observations. Following observations of a lack of normality, the significance of δ$^{13}$C and δ$^{18}$O variation between the two islands (Timor and Alor), and between terrestrial and marine fauna on each island were tested using Mann–Whitney–Wilcoxon tests. The significance of δ$^{13}$C and δ$^{18}$O variation between taxa and between the different periods was tested using Kruskal–Wallis tests. Where significant, these tests were followed by a pairwise Wilcoxon comparison to determine which groups were significantly different from each other. In all cases, the dataset being tested and its size are explicitly stated. All statistical analyses were conducted using the free program R software[74].

All FTIR index values were tested for normality using the Shapiro–Wilk test and histogram observations. Analysis of variance (ANOVA) followed by post hoc Tukey pairwise comparisons were performed for each of the main FTIR indices of enamel apatite (PCI, BAI, BPI, API, and WAMPI—per[45] and defined in Supplementary Table 12) across the sample groups (i.e., 'Fossil Marine Fauna', 'Fossil Terrestrial Fauna', 'Fossil Humans', and 'Modern'), in order to determine statistical differences in enamel crystallinity and structure between fossil and modern samples. All statistical analyses were again conducted using the free program R software[74].

**Reporting summary**. Further information on research design is available in the Nature Research Reporting Summary linked to this article.

## Data availability
All of the data reported in the paper are presented in the main text or in the Supplementary Notes, Tables, Figures, and Data files. The source data underlying Fig. 2 and Supplementary Fig. 16 are provided as a Source Data file. The source data for Fig. 2, as well as Supplementary Data Files 1 and 2, also underlie Figs. 3–6. All other data supporting the findings and interpretations of this study are available in existing publications and the Supplementary Information provided alongside this manuscript. The faunal and human remains sampled from Timor-Leste are curated in the Archaeology collection at the College of Asia and the Pacific, Australian National University, Australia under the site codes J (Asitau Kuru), MK (Matja Kuru), and LH (Lene Hara). These materials are to be returned to Timor-Leste upon construction of a national museum storehouse. The faunal and human remains from Alor are housed in the Department of Archaeology, Universitas Gadjah Mada, Indonesia, under the site codes TBL (Tron Bon Lei) and MP (Makpan). All site codes are followed by suffixes of a single letter denoting the excavation square and a number denoting the spit.

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

## Acknowledgements

For permission to conduct fieldwork, we thank the Secretaria do Estado da Arte e Cultura, Timor-Leste, and the Indonesian Ministry of Research, Technology, and Higher Education (RISTEK) Foreign Research Permit Division (S.O'.C. 1172/FRP/E5/Dit.KI/V/2016). This project was funded by the Max Planck Society, a European Research Council Starter Grant awarded to P.R. (no. 850709), an Australian Research Council Laureate Fellowship awarded to S.O'.C. (FL120100156), and the Australian Research Council Centre of Excellence for Australian Biodiversity and Heritage (CE170100015). We would like to thank the landowners and villagers of Alor and Timor-Leste, staff and students from the Universitas Gadja Mada, Pusat Penelitian Arkeologi Nasional, and Balai Arkeologi Bali for their assistance in the field. We also thank CartoGIS ANU for their assistance.

## Author contributions

P.R., J.L., C.S., S.K., and S.O'C. designed the research; P.R., J.L., J.Z., C.S., S.K., S.S.C., S.H., C.B., S.M., B.F., K.A., and S.O'C. collected the data; P.R., J.L., J.Z., C.S., S.K., S.S.C., S.H., C.B., and S.O'C. analyzed the data; P.R., J.L., J.Z., C.S., S.K., S.S.C., S.H., C.B., S.M., B.F., N.B., M.M., K.A., and S.O'C. wrote the paper.

## Competing interests

The authors declare no competing interests.
