## [Peer Review File · Nature Communications]

Reviewers' Comments:

Reviewer #1:

Remarks to the Author:

The subsistence modes of the seafarers who navigated across Wallacea to colonise Sahul have been debated for decades. This paper makes a significant contribution to this debate using novel stable isotope (mainly carbon) analyses of human and faunal tooth enamel to discriminate marine and terrestrial diet signatures. The headline conclusion of the study is that the earliest people in Wallacea were coastal subsistence specialists with subsistence diversification by later people to include terrestrial forest resources. This is a fascinating conclusion with wide implications.

I enjoyed reading the paper. It is well-written and addresses a globally significant issue. Differentiation of isotopic signals for terrestrial and marine resources is achieved with a large collection of fauna samples. However, there appears to be a mismatch between the framing of the problem and the appropriateness of the dataset used to address it.

The central problem with the paper is the evidentiary basis for the problem as framed and the central conclusion. Only very limited human samples are included in the sample. For Alor, all of the human samples are assigned to the 15ka to 7.4ka year phase, telling us little about the diets of the initial colonists of the region. For Timor, there is a larger number of human samples available. However, only one of the samples dates to before 20ka years ago (dated to 42-38ka). So only one sample in the entire study can tell us anything about an earlier phase of human occupation in Wallacea and even that sample is 20ka+ after when we know people moved through Wallacea in colonising Sahul.

Obviously the sample is the sample. But in this case the sample is not entirely well suited to addressing the question as posed.

Citations in the manuscript very much privilege work published by the authors. It would be appropriate to cite a broader body of work acknowledging other researchers who have made contributions to this problem and in this region.

A few examples:

I was surprised not to see the work of O'Connell and Allen cited given they have contributed widely to ideas about early subsistence in and movement through Wallacea.

The 'savannah corridor' work promoted by Bird and others is not cited even though it is clearly important for the problem posed.

Line 63 – citations on Wallacean voyaging should include work by Norman et al. and Bird et al.

Line 65 – citations on Wallacean diet should include work by O'Connell and Allen

Line 68 – citations on early hominins on Flores should include Morwood et al.

Line 247 – citations on exchange of materials should include Reepmeyer et al.

Line 283 – citations on other hominins in the region should include Morwood et al.

Some minor issues:

Line 60 – reword 'actually ventured challenging adaptive settings'.

Lines 72-74 – delete first sentence of para as it is repeated in the first sentence of the following para.
Line 82 – change 'food resources and they have' to 'food resources. These taxa have been'.
Line 135 – delete 'clear'. Figure 2a clearly shows an overlap in terrestrial and marine $\delta^{13}\text{C}$ signal (which is addressed later in this paragraph).
Line 141 – states that terrestrial fauna from Makpan has a $\delta^{13}\text{C}$ ranging from -19.2 to -3.1. However, the most negative value plotted for Makpan on Figure 5 is -14.5. Check.
Line 159 – change '(as per Louys et al., 2018)' to 22.
Line 271 – change 46 to 45-46.
Line 274 – change 11,47 to 11,46-47.
Line 323 – change 'Organization' to 'Organisation'.
Line 366 – change 'Infra-red' to 'Infrared'.
Line 375 – change '15fossil' to '15 fossil'.
Line 385 – change 'spectroscopy' to 'Spectroscopy'.
Line 523 – change 'hu nting' to 'hunting'.
Figure 1 – main map needs a scale.

Note that I have not reviewed reference formatting completely.

Note that I have not reviewed the supplementary files.

Reviewer #2:

Remarks to the Author:

This is an important paper that reports isotopic data derived from tooth enamel to explain patterns of prehistoric diet in Wallacea, that part of Island Southeast Asia that is between the subcontinents of Sundaland and Sahul. The paper presents quality data based on materials that have been recovered from systematic, controlled excavations at well known (and reported) sites on the islands of Alor and Timor. The Supplementary Materials text provides an excellent summary of these excavations and associated context of human teeth analyzed, associated faunas, etc.

Substantively, the authors have done an excellent job in situating their research problem against the important question of terrestrial vs. marine diet, which is difficult to discern when C4 foods are in the mix. However, the timing and context of these sites make C4 foods a non-issue. The sites included in this study and the few late Pleistocene-Holocene human remains analyzed are backed by solid isotopic data from a diverse range of associated faunas. There are significant differences in both $\delta^{13}\text{C}$ and $\delta^{18}\text{O}$ between the two islands, but the authors focus their attention on carbon isotopic variation, which makes sense given the dietary focus of this paper. It is appropriate, however, that they include all isotopic data generated for the project. There are only a couple of points that are glossed over that may require slight expansion or clarification, and these are raised below (disregard line numbers interspersed in text).

This statement is a bit broad-brushed because there are diverse opinions about late Pleistocene climatic conditions in island Southeast Asia, in particular Wallacea

Lines 92-93. In regions such as Pleistocene Wallacea where tropical forests are thought to have dominated terrestrial environments 27 ...

Would be good to clarify how these target $\delta^{13}\text{C}$ values are produced, in addition to citing the paper. Perhaps a sentence or two that state how these mean values are derived (what are the end members used), etc.

Lines 97-100. Meanwhile, marine producer biomass has higher $\delta^{13}\text{C}$ than all C3 terrestrial plants 28-

29, enabling marine consumers to be distinguished from terrestrial C3 consumers 30. We expect preindustrial humans relying completely on tropical forest, open C3 resources, and marine resources to have $\delta^{13}\text{C}$ values of c. -14‰ , c. -11‰ , and c. -4‰ , respectively 26, 30.

Should clarify that subsample of teeth analyzed for FTIR, not the entire sample
Lines 112-114. The preservation of the analyzed tooth enamel samples was also checked using Fourier Transform Infrared Spectroscopy (FTIR) as per Roberts et al. 26, 33.

Map figure confusing with respect to labels .. data are grouped by 'Alor' and 'Timor' but on map, 'Timor' is not indicated, as only Timor-Leste is indicated.

What follows are small details that can easily be corrected. These include:

Insert 'into'

Lines 59-60. Testing this hypothesis is particularly timely given finds that imply other hominin species may have actually ventured INTO challenging adaptive settings 4, 10.

Missing (=) sign and delete 'df,'

Line 155. faunal groups in terms of $\delta^{18}\text{O}$ on either Timor ($W=1507$, $p=>0.05$) or Alor ($W1375$, df , $p=>0.05$).

Tense Change 'forming' to 'formed'?

Lines 335-336. Subtle variations in $\delta^{13}\text{C}$ and $\delta^{18}\text{O}$ between teeth could occur as a result, particularly for teeth forming during 'weaning' 61.

Fix spelling

Lines 349-350. centrifuging, before 0.1M acetic acid was added for 10 minutes, followed by another three rinses in purified H₂O. Samples were then lysophilized for 24 hours.

Should include $\delta^{13}\text{C}$ and $\delta^{18}\text{O}$ value for internal bovid tooth enamel standard, as you do for USGS44 and MERCK in-house standards

Lines 355-359. 'International Standards (IAEA-603 ($\delta^{13}\text{C} = 2.5$; $\delta^{18}\text{O} = -2.4$); IAEA-CO-8 ($\delta^{13}\text{C} = -5.8$; $\delta^{18}\text{O} = -22.7$); USGS44 ($\delta^{13}\text{C} = -42.2$)) and in-house standard (MERCK ($\delta^{13}\text{C} = -41.3$; $\delta^{18}\text{O} = -14.4$)). Replicate analysis of MERCK standards suggests that machine measurement error is c. $\pm 0.1\text{‰}$ for

$\delta^{13}\text{C}$ and $\pm 0.2\text{‰}$ for $\delta^{18}\text{O}$. Overall measurement precision was studied through the measurement of repeat extracts from a bovid tooth enamel standard ($n = 20$, $\pm 0.2\text{‰}$ for $\delta^{13}\text{C}$ and $\pm 0.3\text{‰}$).

Insert space

Line 375. 14 fossil human, 15fossil terrestrial fauna, and 15 marine fauna samples were subjected to FTIR

Missing hyphen

Line 398, tested using MannWhitney-Wilcoxon tests.

Regarding References, these are inconsistent. Most journal titles are abbreviated, some are not. Some journal abbreviations have periods, some do not. Most article titles are lowercase after first word, some are not. Most book titles are lowercase after first word, some are not. Refs should be reviewed and made consistent in revision.

Italicize

Line 429. 3. Rizal, Y. et al. Last appearance of *Homo erectus* at Ngandong, Java, 117,000-108,000

Fix spelling and lowercase key words after first word of journal article (check refs throughout)

Line 431. 4. Reich, D. et al. Denisova Admixutre and the First Modern Human Dispersals into

Most book titles, you have lowercase after first word of title, this is one exception – there are others). Should be consistent throughout. Also comma after surname.

Line 433. 5. Rabett R.J. Human Adaptation in the Asian Palaeolithic. (Cambridge Univ. Press, Cambridge, 2012).

Spacing between B. and A.

Line 435. 6. Roberts, P. & Stewart, B. A. Defining the 'generalist-specialist' niche for Pleistocene *Homo sapiens*. *Nat Hum Behav* 2, 542–550 (2018).

Fix hyphen for inter-island, and lowercase words after first word in title, also journal title should be abbreviated (*Archaeol. Prospect.*)

Lines 447-449. 12. Kealy, S., Louys, J. & O'Connor, S. Reconstructing Palaeogeography and Inter Island

Visibility in the Wallacean Archipelago During the Likely Period of Sahul Colonization, 65–45 000 Years Ago. *Archaeological Prospection* <https://doi.org/10.1002/arp.1570> (2017).

Uppercase word after colon. Periods after journal abbreviation (check throughout refs.)

Lines 465-466. 19. Roberts, P., Amano, N. Plastic pioneers: hominin biogeography east of the Movius Line during the Late Pleistocene. *Archaeol Res Asia* 17, 181-192 (2019).

Spacing issue

Lines 507-509. 38. Michel, V., Ildefonse, P. & Morin, G. Assessment of archaeological bone and dentine preservation from Lazaret Cave (Middle Pleistocene) in France. *Palaeogeogr Palaeoclimatol Palaeoecol* 126, 109-119 (1996).

Close parenthetical

Lines 510-512. 39. Roche, D. et al. Preservation assessment of Miocene-Pliocene tooth enamel from Tugen Hills (Kenyan Rift Valley - through FTIR, chemical and stable-isotope analyses. *J Archaeol Sci* 37, 1690-1699 (2010).

Abbreviate Journal

Lines 532-534. 48. Boulanger, C. et al. Coastal Subsistence Strategies and Mangrove Swamp Evolution at Bubog I Rockshelter (Ilin Island, Mindoro, Philippines) from the Late Pleistocene to the mid-Holocene. *The Journal of Island and Coastal Archaeology* 14, 584-604 (2019).

Italics needed

Lines 550-552. 56. Aplin, K.P. & Helgen, K.M. Quaternary murid rodents of Timor Part I: New material of *Coryphomys buehleri* Schaub, 1937, and description of a second species of the genus. *Bull Am Mus Nat* 341, 1-80 (2010).

Typos x2

Line 607. stratigraphic and chronological information (Supplemenatry Note 1).

In several figure captions, 'over the top' is confusing and does not really convey anything. Superimposed might be better, but 'white symbols' should be sufficient

Supplemental Information

Spell out Optically Stimulated Luminescence (OSL) at first use (1st paragraph)

Awkward English, phrase 'usually associated with the Holocene' the beads are more than just that

Delete 'to' in Phrase 'possible hiatus just before to the Last Glacial Maximum and a resumption of low'

Delete 2nd 'been' in Phrase "Genus A') have been thus far been recognized from this site, as well as smaller'

... next line

Add 'and' in Phrase 'rodents such as Melomys spp., Komodomys spp., AND Rattus exulans. These taxa are'

Insert comma after however in Phrase 'however excavation units (spits) can be broadly correlated with the units from D at'

At phrase 'deposit was excavated to provide a sample of non-humanly derived fauna for' consider deleting 'ly' off humanly, which would be more appropriate in this context.

THIS Sentence deviates from flow of the supplementary text, so may want to consider how you phrase selection of samples from each site – you do not need to articulate this, as it is a no brainer given the main jist of this paper ... 'Here, we sought to also test the overall dietary reliance of this individual to marine as opposed to terrestrial resources, so a tooth from this individual was sampled, as well as the two overlying burials from Square D, and burials 1 and 2 from Square C.'

Refs. Look okay, except 'Journal of Coastal and Island Archaeology' title needs to be corrected and abbreviated for all entries.

John Krigbaum

Reviewer #3:

Remarks to the Author:

I think this is an excellent paper, and for once I recommend publication as it is. What you proposed makes a great deal of sense, and is well supported by your data and arguments. The model of initial coastal colonisation reliant on marine resources followed by a greater use of inland resources such as giant rats is consistent with our current data, and should encourage future research to investigate this shift of emphasis. I agree entirely with your comments on human adaptability as a colonising creature, and how this talent is amply shown by the colonisation of Wallacea.

Point-by-Point Response to Reviewers

We would like to thank the Reviewers for their highly constructive comments in relation to our manuscript and for taking the time to read it, and the additional files, in detail. We are glad that they all recognised the importance of our data and conclusions. We thank them also for their detailed suggestions to improve our manuscript. We have responded to these on a point-by-point basis below as well as in a word document with tracked changes and believe that they have greatly improved our article.

.....

Reviewer #1 (Remarks to the Author):

The subsistence modes of the seafarers who navigated across Wallacea to colonise Sahul have been debated for decades. This paper makes a significant contribution to this debate using novel stable isotope (mainly carbon) analyses of human and faunal tooth enamel to discriminate marine and terrestrial diet signatures. The headline conclusion of the study is that the earliest people in Wallacea were coastal subsistence specialists with subsistence diversification by later people to include terrestrial forest resources. This is a fascinating conclusion with wide implications. I enjoyed reading the paper. It is well-written and addresses a globally significant issue. Differentiation of isotopic signals for terrestrial and marine resources is achieved with a large collection of fauna samples.

We thank Reviewer 1 in relation to their highly positive comments in relation to the novelty of our approach, data, and conclusions. We are glad that they enjoyed reading the paper and found it to be well-written and well-structured with wider implications. We are also happy that they recognized the significance of our faunal isotopic dataset in terms of its size.

However, there appears to be a mismatch between the framing of the problem and the appropriateness of the dataset used to address it. The central problem with the paper is the evidentiary basis for the problem as framed and the central conclusion. Only very limited human samples are included in the sample. For Alor, all of the human samples are assigned to the 15ka to 7.4ka year phase, telling us little about the diets of the initial colonists of the region. For Timor, there is a larger number of human samples available. However, only one of the samples dates to before 20ka years ago (dated to 42-38ka). So only one sample in the entire study can tell us anything about an earlier phase of human occupation in Wallacea and even that sample is 20ka+ after when we know people moved through Wallacea in colonising Sahul. Obviously the sample is the sample. But in this case the sample is not entirely well suited to addressing the question as posed.

We take the Reviewer's point in relation to the size of our dataset. Unfortunately, as ever with archaeology, particularly for the Late Pleistocene, this is unavoidable, as the Reviewer notes themselves. Our dataset remains the first human and faunal isotopic datasets for Wallacea, and indeed the first Late Pleistocene human isotopic dataset for Island Southeast Asia more widely. As the Reviewer notes, our methodological approach, and our proof of concept through the fauna, is also immensely important for research in the region and will hopefully stimulate more work of this nature across Southeast Asia and Australasia.

Nevertheless, we do believe that our individual from the Asitau Kuru Phase dating to c. 46-29 ka (and more directly to ~42 ka) does speak to the earliest archaeological evidence we have for humans occupying Wallacea. There is no earlier fossil, or material culture, evidence, at present, for humans in this part of the world beyond this site (albeit there are indications of earlier occupation in Australia). We also believe that our comprehensive post-20 ka human sample shows interesting unique trends by comparison to this early phase and represents the only Pleistocene human dataset for Wallacea and Island Southeast Asia, showing the occupation of supposedly 'isolated' interior environments long prior to agriculture.

We, like the Reviewer, hope that further samples in future will allow the additional testing of our conclusions across this part of the world. We hope to have made a significant contribution that provides the groundwork for such work to expand and be prioritized in future.

Citations in the manuscript very much privilege work published by the authors. It would be appropriate to cite a broader body of work acknowledging other researchers who have made contributions to this problem and in this region.

A few examples:

I was surprised not to see the work of O'Connell and Allen cited given they have contributed widely to ideas about early subsistence in and movement through Wallacea. The 'savannah corridor' work promoted by Bird and others is not cited even though it is clearly important for the problem posed.

We thank the Reviewer for pointing this out and suggesting some further reading and citations. We have now cited all of their suggested literature as highlighted below.

Line 63 – citations on Wallacean voyaging should include work by Norman et al. and Bird et al.

We have added both of these references at the relevant position.

Line 65 – citations on Wallacean diet should include work by O'Connell and Allen

We have added this citation.

Line 68 – citations on early hominins on Flores should include Morwood et al.

We have added.

Line 247 – citations on exchange of materials should include Reepmeyer et al.

We have added.

Line 283 – citations on other hominins in the region should include Morwood et al.

We have added.

Some minor issues:

Line 60 – reword 'actually ventured challenging adaptive settings'.

Have changed to:

“ventured into challenging adaptive settings”

Lines 72-74 – delete first sentence of para as it is repeated in the first sentence of the following para.

We have adapted to make more sense. We have tried to make it clear that the end of the preceding paragraph refers to all hominins while the start of the next refers just to our own species.

Line 82 – change ‘food resources and they have’ to ‘food resources. These taxa have been’.

We have changed to:

*“Moreover, there remains the possibility that giant rat taxa, with preferences for closed forest environments and an adult body weight of up to 6 kg, represented significant food resources; **and they have been identified** in early coastal and inland archaeological contexts in Alor and Timor (e.g. ²⁷).”*

Otherwise the sentence would not have quite made sense.

Line 135 – delete ‘clear’. Figure 2a clearly shows an overlap in terrestrial and marine 13C signal (which is addressed later in this paragraph).

Removed as requested.

Line 141 – states that terrestrial fauna from Makpan has a 13C ranging from -19.2 to -3.1. However, the most negative value plotted for Makpan on Figure 5 is -14.5. Check.

Thank you for pointing this out. We have now checked. The point was missing from the Figure 2 and Figure 5 plots which we have now corrected and also adapted the scales of all of the Figures for consistency.

Line 159 – change ‘(as per Louys et al., 2018)’ to 22.

We have changed as requested.

Line 271 – change 46 to 45-46.

We have changed.

Line 274 – change 11,47 to 11,46-47.

We have changed.

Line 323 – change ‘Organization’ to ‘Organisation’.

We have changed.

Line 366 – change ‘Infra-red’ to ‘Infrared’.

We have changed.

Line 375 – change ‘15fossil’ to ‘15 fossil’.

We have changed.

Line 385 – change ‘spectroscopy’ to ‘Spectroscopy’.

We have changed.

Line 523 – change ‘hu nting’ to ‘hunting’.

Changed, thank you for spotting.

Figure 1 – main map needs a scale.

We have now added.

Note that I have not reviewed reference formatting completely.

Note that I have not reviewed the supplementary files.

Reviewer 2 has done so and we have now reviewed both fully based on the Editor’s comments.

Reviewer #2 (Remarks to the Author):

This is an important paper that reports isotopic data derived from tooth enamel to explain patterns of prehistoric diet in Wallacea, that part of Island Southeast Asia that is between the subcontinents of Sundaland and Sahul. The paper presents quality data based on materials that have been recovered from systematic, controlled excavations at well known (and reported) sites on the islands of Alor and Timor. The Supplementary Materials text provides an excellent summary of these excavations and associated context of human teeth analyzed, associated faunas, etc.

We thank Reviewer 2 for their positive comments in relation to our manuscript. We are glad that they appreciate the quality of our data. We thank them also for their positive comments in relation to the thoroughness of our Supplementary Information.

Substantively, the authors have done an excellent job in situating their research problem against the important question of terrestrial vs. marine diet, which is difficult to discern when C4 foods are in the mix. However, the timing and context of these sites

make C4 foods a non-issue. The sites included in this study and the few late Pleistocene-Holocene human remains analyzed are backed by solid isotopic data from a diverse range of associated faunas. There are significant differences in both d13C and d18O between the two islands, but the authors focus their attention on carbon isotopic variation, which makes sense given the dietary focus of this paper. It is appropriate, however, that they include all isotopic data generated for the project. There are only a couple of points that are glossed over that may require slight expansion or clarification, and these are raised below (disregard line numbers interspersed in text).

We are glad the Reviewer likes how we have situated our manuscript and appreciate our data in relation to the isotopic distinction between terrestrial and marine resources. We thank them for their kind comments in relation to our ample baseline dataset and for the way in which we have made our data fully available. We thank Reviewer 2 for pointing out some points for expansion and clarification and have tried our best below to meet their suggestions.

This statement is a bit broad-brushed because there are diverse opinions about late Pleistocene climatic conditions in island Southeast Asia, in particular Wallacea Lines 92-93. In regions such as Pleistocene Wallacea where tropical forests are thought to have dominated terrestrial environments 27 ...

We have changed to:

“In regions such as Pleistocene Wallacea, where some researchers have suggested that tropical forests dominated terrestrial environments³², with grasslands considered largely absent;....”

To temper this statement a little.

Would be good to clarify how these target d13C values are produced, in addition to citing the paper. Perhaps a sentence or two that state how these mean values are derived (what are the end members used), etc.

Lines 97-100. Meanwhile, marine producer biomass has higher $\delta^{13}\text{C}$ than all C3 terrestrial plants 28-29, enabling marine consumers to be distinguished from terrestrial C3 consumers 30. We expect preindustrial humans relying completely on tropical forest, open C3 resources, and marine resources to have $\delta^{13}\text{C}$ values of c. -14‰, c. -11‰, and c. -4‰, respectively 26, 30.

We have adjusted to read as follows:

“Based on research done in East Africa³⁰, Sri Lanka³¹, and Japan³⁵, including extensive modern studies³⁰, we expect pre-industrial humans relying completely on tropical forest, open C₃ resources, and marine resources to have tooth enamel $\delta^{13}\text{C}$ values of c. -14‰, c. -11‰, and c. -4‰, respectively.”

Should clarify that subsample of teeth analyzed for FTIR, not the entire sample Lines 112-114. The preservation of the analyzed tooth enamel samples was also checked using Fourier Transform Infrared Spectroscopy (FTIR) as per Roberts et al. 26, 33.

We have changed to:

“The preservation of a sub-section of the analysed tooth enamel samples was also checked using Fourier Transform Infrared Spectroscopy (FTIR) as per Roberts et al. ^{31, 38}.”

Map figure confusing with respect to labels .. data are grouped by ‘Alor’ and ‘Timor’ but on map, ‘Timor’ is not indicated, as only Timor-Leste is indicated.

We have changed to Timor throughout as this defines the island. We have, however, indicated on a new version of Figure 1 that the island of Timor is a part of the territory of Timor-Leste today.

What follows are small details that can easily be corrected. These include:

Insert ‘into’

Lines 59-60. Testing this hypothesis is particularly timely given finds that imply other hominin species may have actually ventured INTO challenging adaptive settings 4, 10.

We have inserted as per Reviewer 1’s suggestion above as well.

Missing (=) sign and delete ‘df,’

Line 155. faunal groups in terms of $\delta^{18}\text{O}$ on either Timor ($W=1507$, $p=>0.05$) or Alor ($W1375$, df , $p=>0.05$).

We have changed accordingly.

Tense Change ‘forming’ to ‘formed’?

Lines 335-336. Subtle variations in $\delta^{13}\text{C}$ and $\delta^{18}\text{O}$ between teeth could occur as a result, particularly for teeth forming during ‘weaning’ 61.

We have modified to the correct tense.

Fix spelling

Lines 349-350. centrifuging, before 0.1M acetic acid was added for 10 minutes, followed by another three rinses in purified H₂O. Samples were then lysophilized for 24 hours.

Corrected.

Should include $\delta^{13}\text{C}$ and $\delta^{18}\text{O}$ value for internal bovid tooth enamel standard, as you do for USGS44 and MERCK in-house standards

Lines 355-359. ‘International Standards (IAEA-603 ($\delta^{13}\text{C} = 2.5$; $\delta^{18}\text{O} = -2.4$); IAEA-CO-8 ($\delta^{13}\text{C} = -5.8$; $\delta^{18}\text{O} = -22.7$); USGS44 ($\delta^{13}\text{C} = -42.2$)) and in-house standard (MERCK ($\delta^{13}\text{C} = -41.3$; $\delta^{18}\text{O} = -14.4$)). Replicate analysis of MERCK standards suggests that machine measurement error is c. $\pm 0.1\%$ for $\delta^{13}\text{C}$ and $\pm 0.2\%$ for $\delta^{18}\text{O}$. Overall measurement precision was studied through the measurement of repeat extracts from a bovid tooth enamel standard ($n = 20$, $\pm 0.2\%$ for $\delta^{13}\text{C}$ and $\pm 0.3\%$.)’

We have added.

Insert space

Line 375. 14 fossil human, 15fossil terrestrial fauna, and 15 marine fauna samples were

subjected to FTIR

We have inserted, thank you for spotting.

Missing hyphen

Line 398, tested using MannWhitney-Wilcoxon tests.

We have added.

Regarding References, these are inconsistent. Most journal titles are abbreviated, some are not. Some journal abbreviations have periods, some do not. Most article titles are lowercase after first word, some are not. Most book titles are lowercase after first word, some are not. Refs should be reviewed and made consistent in revision.

We have now reviewed all of our references and made them consistent in terms of journal titles, journal abbreviation formats, and article and book titles. We thank the Reviewer for going through these in detail below.

Italicize

Line 429. 3. Rizal, Y. et al. Last appearance of Homo erectus at Ngandong, Java, 117,000-108,000

Changed

Fix spelling and lowercase key words after first word of journal article (check refs throughout)

Line 431. 4. Reich, D. et al. Denisova Admixture and the First Modern Human Dispersals into

**Most book titles, you have lowercase after first word of title, this is one exception – there are others). Should be consistent throughout. Also comma after surname.
Line 433. 5. Rabett R.J. Human Adaptation in the Asian Palaeolithic. (Cambridge Univ. Press, Cambridge, 2012).**

Changed

Spacing between B. and A.

Line 435. 6. Roberts, P. & Stewart, B. A. Defining the ‘generalist-specialist’ niche for Pleistocene Homo sapiens. Nat Hum Behav 2, 542–550 (2018).

Fixed.

Fix hyphen for inter-island, and lowercase words after first word in title, also journal title should be abbreviated (Archaeol. Prospect.)

Lines 447-449. 12. Kealy, S., Louys, J. & O’Connor, S. Reconstructing Palaeogeography and Inter Island

Visibility in the Wallacean Archipelago During the Likely Period of Sahul Colonization, 65–45 000 Years Ago. Archaeological

Prospection <https://doi.org/10.1002/arp.1570> (2017).

Fixed.

Uppercase word after colon. Periods after journal abbreviation (check throughout refs.)
Lines 465-466. 19. Roberts, P., Amano, N. Plastic pioneers: hominin biogeography east of the Movius Line during the Late Pleistocene. *Archaeol Res Asia* 17, 181-192 (2019).

Fixed.

Spacing issue

Lines 507-509. 38. Michel, V., Ildefonse, P. & Morin, G. Assessment of archaeological bone and dentine preservation from Lazaret Cave (Middle Pleistocene) in France. *Palaeogeogr Palaeoclimatol Palaeoecol* 126, 109-119 (1996).

Fixed.

Close parenthetical

Lines 510-512. 39. Roche, D. et al. Preservation assessment of Miocene-Pliocene tooth enamel from Tugen Hills (Kenyan Rift Valley - through FTIR, chemical and stable-isotope analyses. *J Archaeol Sci* 37, 1690-1699 (2010).

Fixed.

Abbreviate Journal

Lines 532-534. 48. Boulanger, C. et al. Coastal Subsistence Strategies and Mangrove Swamp Evolution at Bubog I Rockshelter (Ilin Island, Mindoro, Philippines) from the Late Pleistocene to the mid-Holocene. *The Journal of Island and Coastal Archaeology* 14, 584-604 (2019).

Abbreviated.

Italics needed

Lines 550-552. 56. Aplin, K.P. & Helgen, K.M. Quaternary murid rodents of Timor Part I: New material of *Coryphomys buehleri* Schaub, 1937, and description of a second species of the genus. *Bull Am Mus Nat* 341, 1-80 (2010).

Italics added.

Typos x2

Line 607. stratigraphic and chronological information (Supplemenatry Note 1).

Fixed.

In several figure captions, ‘over the top’ is confusing and does not really convey anything. Superimposed might be better, but ‘white symbols’ should be sufficient

We have changed as suggested.

Supplemental Information

Spell out Optically Stimulated Luminescence (OSL) at first use (1st paragraph)

We have now spelled this out at its first mention.

Awkward English, phrase ‘usually associated with the Holocene’ the beads are more than just that

We have removed this phrase.

Delete ‘to’ in Phrase ‘possible hiatus just before to the Last Glacial Maximum and a resumption of low’

Deleted.

Delete 2nd ‘been’ in Phrase ‘‘Genus A’) have been thus far been recognized from this site, as well as smaller’

Deleted.

... next line

Add ‘and’ in Phrase ‘rodents such as Melomys spp., Komodomys spp., AND Rattus exulans. These taxa are’

Added.

Insert comma after however in Phrase ‘however excavation units (spits) can be broadly correlated with the units from D at’

Inserted.

At phrase ‘deposit was excavated to provide a sample of non-humanly derived fauna for’ consider deleting ‘ly’ off humanly, which would be more appropriate in this context.

Deleted.

THIS Sentence deviates from flow of the supplementary text, so may want to consider how you phrase selection of samples from each site – you do not need to articulate this, as it is a no brainer given the main jist of this paper ... ‘Here, we sought to also test the overall dietary reliance of this individual to marine as opposed to terrestrial resources, so a tooth from this individual was sampled, as well as the two overlying burials from Square D, and burials 1 and 2 from Square C.’

Changed as suggested. Thank you for reading the Supplementary docs in detail!

Refs. Look okay, except ‘Journal of Coastal and Island Archaeology’ title needs to be corrected and abbreviated for all entries.

We have corrected and reviewed as with the main text references.

Reviewer #3 (Remarks to the Author):

I think this is an excellent paper, and for once I recommend publication as it is. What you proposed makes a great deal of sense, and is well supported by your data and arguments. The model of initial coastal colonisation reliant on marine resources followed by a greater use of inland resources such as giant rats is consistent with our current data, and should encourage future research to investigate this shift of emphasis. I agree entirely with your comments on human adaptability as a colonising creature, and how this talent is amply shown by the colonisation of Wallacea.

We thank Reviewer 3 for agreeing with Reviewers 1 and 2 that this is an excellent paper. We are glad they find that our data well supports our arguments. We likewise hope that our paper will stimulate future research on direct assessments of hominin adaptations and the degree to which our species differed from other hominins in its adaptability and process of colonization.